# Dielectric Loss and Electrical Conductivity Behaviors of Epoxy Composites Containing Semiconducting ZnO Varistor Particles

**DOI:** 10.3390/molecules27186067

**Published:** 2022-09-17

**Authors:** Li Lei, Chaoxin Chen, Haoran Nie, Xudong Wu, Daniel Q. Tan

**Affiliations:** 1Department of Materials Science and Engineering, Guangdong Technion-Israel Institute of Technology, 241 Daxue Road, Shantou 515063, China; 2Department of Materials Science and Engineering, Technion-Israel Institute of Technology, Haifa 3200003, Israel; 3Guangdong Provincial Key Laboratory of Materials and Technology for Energy Conversion, 241 Daxue Road, Shantou 515063, China

**Keywords:** ZnO varistor, dielectric permittivity, composites, particle interconnect, epoxy impregnation

## Abstract

Polymer nanodielectrics render a great material platform for exhibiting the intrinsic nature of incorporated particles, particularly semiconducting types, and their interfaces with the polymer matrix. Incorporating the oxide fillers with higher loading percentages (>40 vol%) encounters particular challenges in terms of dispersion, homogeneous distribution, and porosity from the process. This work investigated the dielectric loss and electrical conduction behaviors of composites containing semiconducting ZnO varistor particles of various concentrations using the epoxy impregnation method. The ZnO varistor particles increased the dielectric permittivity, loss, and electrical conductivity of the epoxy composites into three different regimes (0–50 vol%, 50–70 vol%, 70–100 vol%), particularly under an electric bias field or at higher temperatures. For lower loading fractions below 50 vol%, the dielectric responses are dominated by the insulating epoxy matrix. When loading fractions are between 50 and 70 vol%, the dielectric and electric responses are mostly associated with the semiconducting interfaces of ZnO varistor particles and ZnO–epoxy. At above 70 vol%, the apparent increase in the dielectric loss and conductivity is primarily associated with the conducting ZnO core forming the interconnected channels of electric conduction. The foam-agent-assisted ZnO varistor particle framework appears to be a better way of fabricating composites of filler loading above 80 vol%. A physical model using an equivalent capacitor, diode, and resistor in the epoxy composites was proposed to explain the different property behaviors.

## 1. Introduction

A nanodielectric polymer is a composite with a polymer serving as the matrix and inorganic particles as the fillers. It was intensively explored for the purpose of application in energy storage capacitors, electrical insulation, and voltage surge suppressing such as in metal oxide varistors. Designing polymer composites containing a higher volume percentage of the fillers can expand the metal oxide varistor product and improve the nonlinear electrical behavior of composite materials for electrical system protection. Since 2000, it has been extensively investigated in the dielectric arena because of its potential for better dielectric materials with higher dielectric permittivity, lower dielectric loss, higher dielectric strength, and miniaturization. It is designed to synergize the advantages of both inorganic fillers, such as high dielectric permittivity or thermal conductivity, and organic matrices, such as high dielectric strength or flexibility for electrical insulation or film capacitor purposes. By changing the types, aspect ratio, morphologies, or volume percentage of fillers, one may probe the filler’s contribution, matrix–filler interface, and filler distribution. Amongst all, the effect of filler loading percentage has been widely researched at a high-volume fraction (>10 vol%) of high-permittivity fillers (K > 1000) to raise the dielectric permittivity of a polymer matrix [1,2,3]. On the opposite side, the effects of very low filler volume loading (<10 vol%) in increasing the composite’s energy density are also explored [4,5,6]. However, it is not easy work to do when the loading fraction reaches percolation level or exceeds 30 vol% of fillers because of the mixing difficulty, porosity elimination, and filler interconnection. One can barely find where the filler’s volume loading exceeds 50 vol% in the publication. Alternative methods such as the impregnation of polymer solution into particle network present to be effective in fulfilling the composite fabrication and properties [6].

In addition, the dielectric permittivity and electrical resistivity closely depend on the electrical nature of the fillers. Semiconducting and conducting fillers generally cause a significant increase in dielectric permittivity, loss, and electrical conductivity (reduced resistivity), as shown in Figure 1. Unlike insulating and conducting fillers, semiconducting fillers cause not only certain leakage currents but also form semiconducting interfaces subjecting to the electrical tunneling-like behavior of diodes [7,8,9]. It requires well-designed composite materials, in-depth and comprehensive models, new characterization tools, and measuring methods [10,11]. A varistor is a material structure sensitive to the applied electrical field and exhibits a nonlinear increase in electrical conduction with an increasing electric field. The nonlinear relationship between the voltage and electrical current is a crucial phenomenon. It has been widely utilized in metal oxide varistors to protect against high voltage or current transients in modern electrical power systems and electronics [12,13]. Compared with a metal oxide varistor (MOV), the polymer-based nanocomposites provide another design dimension and advantage in terms of high operation voltage excellence, smaller size, higher flexibility, and equivalent nonlinearity [11,12,13]. Designing polymer composites containing a higher volume percentage of the fillers can expand the metal oxide varistor product and readily change the nonlinear behavior for desired over-voltage protection needs. This composite type requires a higher filler fraction close to the percolation level to trigger the nonlinear responses.

Furthermore, polymer-based variable resistor behavior is one of the interesting categories to examine, but its investigation at high filler concentration is limited because of the difficulty of composite processing. Despite a great deal of work addressing the insulating fillers for dielectric capacitor application and highly conducting fillers for thermistor application, there is limited knowledge about the detailed dielectric and electrical responses of composites containing semiconducting fillers such as ZnO particles in the middle part of Figure 1 [13,14,15,16,17,18,19,20]. Despite some composites filled with ZnO particles, no publications have used ZnO varistor particles (ZnO core with additive oxide shell) as the fillers so far. Among many polymer and filler choices, the insulating epoxy and semiconducting ZnO become excellent model materials for their wide electrical power system applications [15,16,17]. Both conducting phase (ZnO single crystal or particle) and ZnO–ZnO interfaces and ZnO varistor–epoxy interfaces of the semiconducting phase would exist. Yet, the two phases behave under electric fields at various temperatures, and their underlined mechanisms are rarely known. In this work, we designed sub-micron ZnO varistor particles by calcining nano-sized ZnO particles with other additive oxides desirable for varistor behavior. In addition, how semiconducting fillers such as ZnO interact with the host polymer is rarely understood. We developed the method of composite fabrication and made it possible to understand the nonlinear electrical behavior. The dielectric properties of epoxy–ZnO varistor composites with different higher vol% ZnO varistor particles exhibited nonlinear characteristics under biased electric fields or at higher temperatures. One may determine the percolation threshold by measuring the dielectric and electrical responses of the nanocomposites fabricated using the disclosed method.

## 2. Experimental Section

### 2.1. Synthesis and Sintering of ZnO Varistor Particles

All precursor powders are oxides of nanometers ranging from 10 to 100 nm purchased from Aladdin, China. The 30 nm ZnO powders are used as the core particles mixed with 5.24 wt% of Bi_2_O_3_, 4.90 wt% of Sb_2_O_3,_ and other additives, which are used to form the satellite particles around ZnO to realize the nonlinear varistor behaviors [3,9]. Detailed compositions are given in Table 1. The amount of MnO_2_, Co_3_O_4_, Cr_2_O_3_, NiO, and SiO_2_ are adjustable to improve the grain growth and nonlinearity of the sintered ceramics. The powder was then milled, dried, and de-agglomerated through a sieve of 120 mesh. Then, the calcination was conducted at 750 °C for 2 h in the air, followed by a second round of ball milling, drying, and de-agglomeration. The dense ceramic samples are produced by sintering the particle-varistor-based compacts at 1150 °C for 3 h, which are pasted with silver as reference varistor samples.

### 2.2. Processing of Epoxy-Based Nanocomposites

To produce highly loaded nanoparticles into a polymer matrix, we used two methods to process nanocomposites: the conventional mixing and the epoxy impregnation of ceramic foams (Figure 2). Before nanocomposite fabrication, the epoxy EPON 828 was degassed at 70 °C for 12 h in a vacuum oven to remove the trapped moisture. After that, the methyl-tetrahydro phthalic anhydride curing agent and 2-ethyl-4-methylimidazole (2E4MZ) catalytic agent were poured into the EPON 828 and stirred vigorously at 70 °C for 0.5 h. The weight ratio used in the present study for EPON 828, the curing agent, and the catalyst was 40:32:0.4. Then, the ZnO varistor powder from calcination was ball milled with the EPON 828 mixture to achieve uniform distribution. The composites with different volume loadings are produced by adjusting the weight ratio between the calcined powder and EPON 828 mixture (20, 30, 45, 50 vol%). They are then cured in 180 °C ovens for 2 h. Finally, the partially finished bulk samples were polished to 1 mm thick discs. The composite discs were then sputter-coated with gold, followed by silver paste coating for subsequent tests.

On the other hand, the porous ceramic matrices are fabricated as the epoxy/nanoparticle composite skeleton by mixing the calcined varistor powders with a forming agent PMMA (Polymethyl Methacrylate). Two different-size PMMA particles, PL-20 and PL-100, enable the pore diameters of 20 μm and 100 μm, respectively. These calcined powders, PL-20/PL-100 particles, and 5 wt% binder agent (PVA) were thoroughly mixed by ball milling using deionized (DI) water, followed by drying, grinding, and slight compacting. The compacted discs were then heated to 750 °C to completely remove the PVA binder, moisture, and PMMA foaming agent particles. By adjusting the weight ratio between the calcined varistor powder and PMMA particles, foam ceramics with 45, 50, 60, 70, and 80 vol% ZnO varistor particles can be fabricated.

The prepared ceramic foams were then immersed into the EPON 828 mixture and impregnated with epoxy in a vacuum oven at 100 °C for 12 h. Then, the oven temperature was raised to 180 °C for 2 h to ensure the complete reaction. The two-step process enables degasification, epoxy impregnation, and curing simultaneously for dense composites. Lastly, the partially finished samples are polished to 1 mm thick discs for electrodes with gold and silver paste for subsequent dielectric tests.

### 2.3. Characterizations of Ceramics and Composites

An X-ray diffractometer examined the calcined varistor powders. In addition, the microstructures of the powders and the foam structure were imaged by a scanning electron microscope (SEM, Gemini SEM 450, Zeiss, Montabaur, Germany). The dielectric constant (permittivity), dielectric loss as the unitless intrinsic property, and electrical conductivity of the samples were measured and automatically calculated by a broadband dielectric spectrometer (Concept 41, Novocontrol Technologies GmbH & Co. KG, Montabaur, Germany) using a frequency ranging from 1 Hz to 10^4 Hz. Moreover, a high-voltage generator unit (HVB4000, Novocontrol Technologies GmbH & Co., Montabaur, Germany.) and a temperature control system (Novocool, Novocontrol Technologies GmbH & Co., Montabaur, Germany) were used to measure the dielectric responses under different DC bias voltages and different temperatures, respectively. This work’s electric conductivity and dielectric properties are sensitive to electrical fields and temperatures due to the semiconducting fillers. They are measured under six bias fields (1, 100, 500, 1000, 1500, and 1900 V/mm) and seven temperatures (25, 50, 75, 100, 125, 150, and 175 °C).

## 3. Results and Discussion

### 3.1. Synthesized ZnO Varistor Powders

The first step of composite fabrication is the synthesis of well-designed ZnO varistor particles (not regular ZnO particles). The ZnO core, the additive oxides, and the reacted phases are determined by the XRD spectra (Figure 3a). It can be confirmed that the calcined ZnO-based powders comprise the Spinel phase (mainly Zn_7_Sb_2_O_12_) and Pyrochlore phase (mainly Zn_2_Bi_3_Sb_3_O_14_) and the ZnO core phase. In addition, a small amount of the Bi_2_O_3_ remains unreacted with other additives and acts as the sintering agent due to its lower melting temperature. All of this is observable on XRD spectra even after sintering at 1150 °C (data not shown here).

The morphology of the additive oxides added to the ZnO particles after calcination and ball milling is inspected using an SEM technique. The majority of additive nanoparticles (30–100 nm) are distributed around the surface of the grown ZnO particles (300–500 nm), as shown in Figure 3b. Point 1 in Figure 3b marks the ZnO grain (single particle, single crystal) confirmed with the EDS. In contrast, Point 2 represents the complex combination of the Spinel and Pyrochlore phases, which are desirable interfaces or barriers for electrical charge to overcome while in an electric field. They will also evolve to be the primary grain boundary phase in a sintered ceramic required to exhibit the characteristics of a metal oxide ceramic varistor. The image demonstrates the essential feature of varistor-like particle morphology in the mixed powder after calcination at 750 °C. The larger ZnO particles act as the ‘core’, and Spinel and Pyrochlore particles act as the decoration on the ZnO core. The core’s shell-like particles and compositions utilized in this work give rise to typical ZnO varistor properties. Nanoparticles are desirable for homogeneous composite mixing and nonlinear property enhancement. However, it is not easy to control the ZnO particle size after calcination that converts the raw nanomaterials to ZnO varistor particles. These particles in the size range of 200 to 500 nm are well accepted in this investigation.

Figure 3c shows the current–voltage relation of the sintered ZnO varistor. The newly developed ZnO varistor indicates a significantly higher varistor field (~630 kV/mm) than the commercial ZnO varistor (~430 kV/mm), rendering an advantage of a thinner device. These results confirm the ZnO varistor particles developed using the calcination profile have the useful varistor feature that can be used for polymer composite fabrication.

### 3.2. Microstructures of Nanodielectric Composites

The microstructures of the ZnO varistor-filled composite that use a foam agent are examined using an SEM technique. Figure 4a shows the random and uniform distributions of pores in the ceramic compacts before epoxy impregnation. The pore size varies from 20 μm to 100 μm. The observation means the processing of the foam agent is under reasonable control. This variation can be attributed to the high-speed ball milling, which results in the breakup of the big PMMA foam agent particles.

The microstructural characterization of epoxy-impregnated foam ceramic was further examined using the SEM by polishing the cured composite samples. Figure 4b shows a different morphology where the homogeneous distribution of ZnO varistor particles appears in the epoxy matrix (45 vol% loading). Since the grinding force makes the topographic change undetectable by the SE signals, a back-scattered electron signal was detected to show the atomic weight contrast between the epoxy-filled area and the ZnO area in the foamed ceramic matrix. EDS inspection confirms the compositional difference between the epoxy and ceramic matrix. Figure 4c,d show the distinct Zn peak for the composite and zero Zn peak but high-intensity peaks for O, C, and H for the epoxy-filled pores. These results confirm that the epoxy was successfully impregnated into the pores in the foam ceramics. The random distribution of ZnO particles indicates the uniformity of ZnO-based foam ceramic. In addition, BET gas absorption measurements were conducted, which confirmed insignificant porosity in the filled foam ceramic composites.

### 3.3. Dielectric and Electrical Properties of Nanocomposites

#### 3.3.1. Temperature and Frequency Dependence for Semiconducting Behavior Exhibition

The temperature dependence of a dielectric material depends on the polarization mechanism, which can help show the sources of dielectric loss and conduction loss. For the semiconducting fillers, a higher conductivity (lower resistance) is expected at higher temperatures. Dielectric loss increases at more elevated temperatures due to larger polarization from thermally agitated dipoles in the dielectric media. The temperature dependence of dielectric responses and conductivity of the epoxy composites containing various volume percentages of ZnO varistor particles are generally measured using a frequency range of 1–100 kHz at a weak electric field. Figure 5 shows that the dielectric permittivity at 1 kHz (a common frequency for dielectric evaluation) under a bias of 1000 V/mm increases gradually from 9 (epoxy only) to 100 (ceramic only) when increasing the loading fraction of ZnO particles [20,21,22]. However, most compositions are nearly independent of temperatures unless ZnO varistor content reaches 60 vol%. Correspondingly, the dielectric loss increases with an increase in temperature above 125 ºC and becomes more significant for compositions with 60 and 70 vol% ZnO varistor particles. Their dielectric loss is two orders of magnitudes higher than that of epoxy (~0.01) [23]. The vast increase can be associated with the contribution of the larger polarizations of the ZnO–ZnO interfaces and ZnO–epoxy interfaces. For compositions from 80 vol% to 100% ZnO varistor (ceramic sample), the dielectric loss is even higher than 100%, a clear indication of the conductive nature of the samples. Electric conductivity measurements reveal more direct evidence of the transition from semiconducting predominance to conducting. The increase in conductivity with the increase in temperature in Figure 5c is the semiconductive characteristic of increasing the ZnO particle concentration, which rises nearly three orders of magnitude from 10^−10^ S/m to 10^−7^ S/m.

In contrast, the pure ceramic sample of the same varistor composition exhibits a decreased conductivity with increasing temperature, a conductive characteristic. These phenomena suggest that the critical composition needed to form the conducting path via conducting ZnO particle inside epoxy is approximately 70–80 vol% ZnO. Below that, the interfaces predominate the dielectric loss.

The complex dielectric responses as a function of frequencies were also measured to better outline the composite effect. Figure 6a shows the pronounced dependence of dielectric permittivity on the frequency for the composites containing ZnO varistor particles higher than 50 vol%. The magnitude of decrease in dielectric permittivity at higher frequencies becomes lower. For the lower loading fraction, such as 20 vol%, the frequency dependence is unmeasurable. Similarly, the dielectric loss in the composites with lower loading fraction is on the lower level of the epoxy host. Increasing the filler concentration results in a remarkable increase and frequency dependence (Figure 6b), indicating the increased filler contribution and its semiconducting effect.

#### 3.3.2. Electric Bias Effect for Conducting Behavior Exhibition

Complex dielectric responses of the composite containing electric-field-dependent ZnO varistors are rarely understood. Because the conductivity in this material system consists of DC conduction (leakage current) in the ZnO particles and dielectric loss related to the interfaces and filler themselves, dielectric and conductivity measurements under the bias electric field will help differentiate their contribution. The electric field dependences of both composites and pure ceramic as shown in Figure 7 at 1 kHz and 125 °C. The composites follow the expected general rules of the increase in dielectric permittivity with the addition of high-permittivity ZnO fillers. The dielectric and electric responses fall into three compositional regimes. The first regime contains ZnO fillers of below 50 vol%. These composites maintain a low conductivity, permittivity, and loss factor, weakly responding to the increased bias fields. The reason for this is that the epoxy has the dominant insulating property, and the added ZnO varistor particles stay separated in the host epoxy, giving rise to negligible contribution and thus little dielectric loss. All the composites remain good insulators withstanding high electric fields.

The second regime contains compositions between 50 vol% and 70 vol% of ZnO particles, exhibiting significant distinction from the first regime. The conductivity is ten times higher, the permittivity is five times higher, and bias dependence also occurs. This is because a sufficient amount of fillers form specific interconnections partially contributing to the composite’s properties. The increase in these dielectric and electric properties is more associated with the interfaces of ZnO–ZnO particles and ZnO–epoxy.

The third regime contains ZnO particles above 70 vol% and shows an extremely high dielectric loss (above 1) under higher bias fields, which indicates the metallic-type conducting nature (Figure 7b,c). The dielectric properties of the compositions are also ZnO filler dominant instead of epoxy dominant since the ZnO varistor particles are interconnected or close. An applied bias field can easily break down the interfacial barriers at such a high loading concentration leading to electrical conduction. Figure 7d shows the conductivity as a function of ZnO varistor particle loading, which marks the transition to conduction under a bias of 1000 kV/mm. The SEM image of the ZnO varistor particles of 80 vol% is shown in the inset displaying the interconnection of the particles in such a high filler concentration. Comparatively, the direct mixing of ZnO particles with epoxy fails to deliver the increased dielectric loss and conductivity up to 50 vol% loading concerning the pure epoxy, which suggests the isolation of primary ZnO particles in the composites.

#### 3.3.3. Analysis of Particle Interconnection at Higher Concentrations

To provide a more pertinent analysis of the significant change in dielectric loss and conductivity previously, the authors proposed a schematic model in Figure 8. The ZnO varistor particles embedded in the epoxy are equivalent to a ZnO core resistor, and additive-phase diodes on both sides of the ZnO core are in contact with epoxy dielectric (two equivalent capacitors), provided an electric field is applied vertically. A very high bias field or temperature is required to overcome two diode barriers (or tunneling) to induce local leakage or dielectric loss. Because of the presence of an epoxy capacitor, no pronounced leakage is expected unless the ZnO varistor particles are highly loaded. It is common to obtain certain ZnO varistor particles with no conformal decoration of additive oxides, as shown by the image on the right of Figure 8a. The bias field will easily overcome one diode barrier and cause higher leakage and dielectric loss. The ZnO particle interconnection pre-set by the foaming agent and binders is evident because the compact fabricated under a high uniaxial press holds more ZnO varistor particles together at a higher probability than the direct mixing of epoxy and particles. The interconnected ZnO varistor framework will respond more sensitively to the applied electric field according to a dielectric material’s electric field screening principle.

It can be conjectured that the ZnO varistor particles below 50 vol% are randomly dispersed in the matrix as isolated from one another. No conducting paths exist, and very low dielectric loss and conductivity occur. The ZnO–epoxy interaction occurs through the epoxy’s molecular chain bonding to the ZnO varistor particle surface, which is generally believed to form an interface in a three-layer particle model [11]. When the filler fraction increases to 60 vol%, some of the particles become closer, resulting in more local clusters and interfaces. Due to their increased contribution, there appears to be higher dielectric loss and conductivity. When further increasing the ZnO particles to 70 vol% or higher, the proximity of ZnO particles subjects to tunneling through the conduction paths among ZnO particles, thus giving rise to a vast conductivity. The electrically lossy behavior is accompanied by the higher permittivity of the composites, as expected.

## 4. Conclusions

Polymer composites containing a higher volume percentage of fillers can be designed for energy storage capacitors using high-dielectric-permittivity films, electrical cable insulation exhibiting semiconducting behavior, and voltage surge suppressors such as nonlinear metal oxide varistors. This work developed ZnO varistor particles by controlling the additive oxides and the calcination profile, which were processed into the epoxy composites for improved nonlinear electrical behavior. After compaction, the highly loaded composites were successfully made feasible via epoxy impregnation into the foaming-agent-assisted ZnO varistor framework. The composites with oxide fillers of >80 vol% were shown to achieve dielectric permittivity as high as 100 and differentiation of electrical conductivity due to conducting ZnO particles and dielectric loss caused by the epoxy–ZnO varistor interfaces in the composites. In the temperature ranging from 25 °C to 175 °C and a bias electric field ranging from 1 V/mm to 1900 V/mm, three regimes of filler loading concentration appear based on the dielectric loss and conductivity behaviors of the composites. Below 50 vol% fillers, the composites behave as insulators, while between 50 and 70 vol%, the higher dielectric loss is associated with the semiconducting nature of the varistor additives around the ZnO core particles. Above 70 vol%, the composites are dominated by the conducting nature of ZnO core particles. Their interconnection or proximity forms the conduction paths contributing to enhanced electric conductivity. A simple model on equivalent capacitor, diode, and resistor was used to explain the three different types of property changes. The vol% dependence of dielectric and electrical conductivity for highly loaded composites is more apparent under a bias field or higher temperatures.

## Figures and Tables

**Figure 1 molecules-27-06067-f001:**
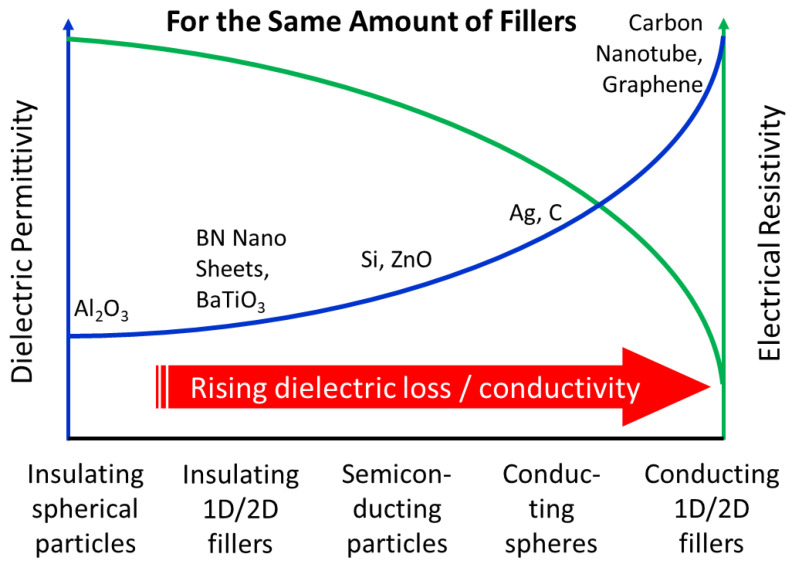
The schematic trend of dielectric response and electric resistivity with increasing the electric conductivity and aspect ratios of the fillers.

**Figure 2 molecules-27-06067-f002:**
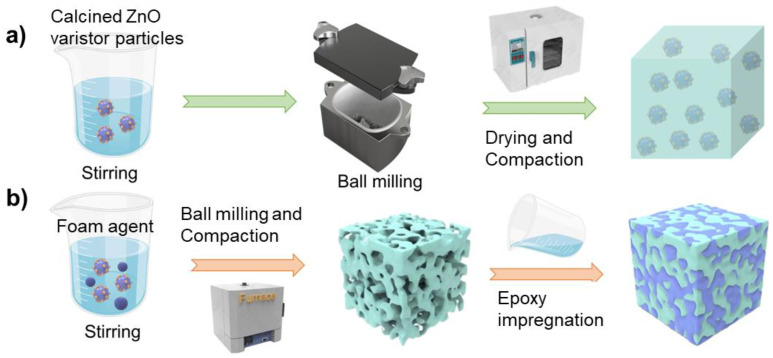
(**a**) Conventional mixing fabrication of composites; (**b**) Impregnated form ceramic fabrication.

**Figure 3 molecules-27-06067-f003:**
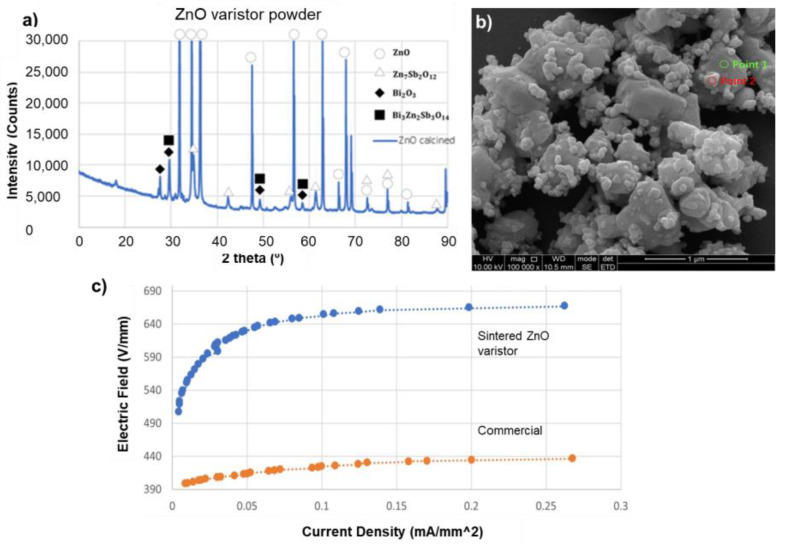
(**a**) XRD spectra of the synthesized powders containing ZnO and oxide additives; (**b**) SEM image of the synthesized powder showing a submicron ZnO particle decorated with nano-sized additive particles; (**c**) I–V curves of the sintered ceramics made of ZnO varistor calcined at 750 °C and a commercial metal oxide varistor.

**Figure 4 molecules-27-06067-f004:**
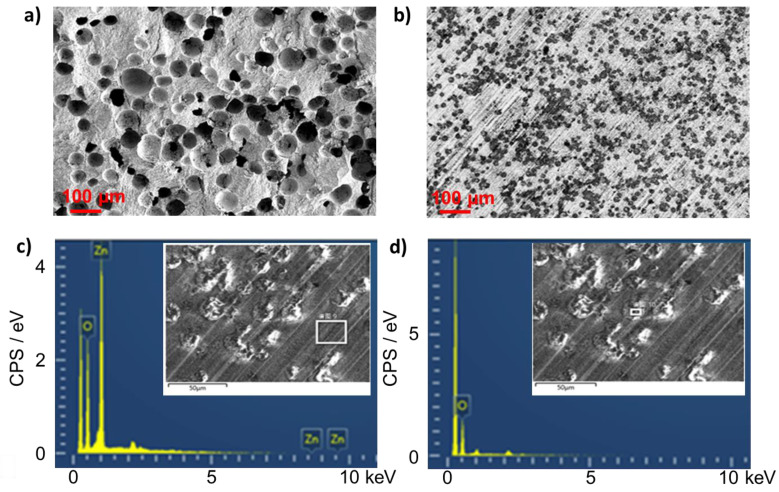
(**a**) The SEM image of the cross-section of the foam ceramic structure without impregnation, (**b**) BSD signal on SEM image of epoxy impregnated foam ceramic, (**c**) EDS of the ZnO area in the ceramic matrix, and (**d**) EDS of the epoxy-filled pore area.

**Figure 5 molecules-27-06067-f005:**
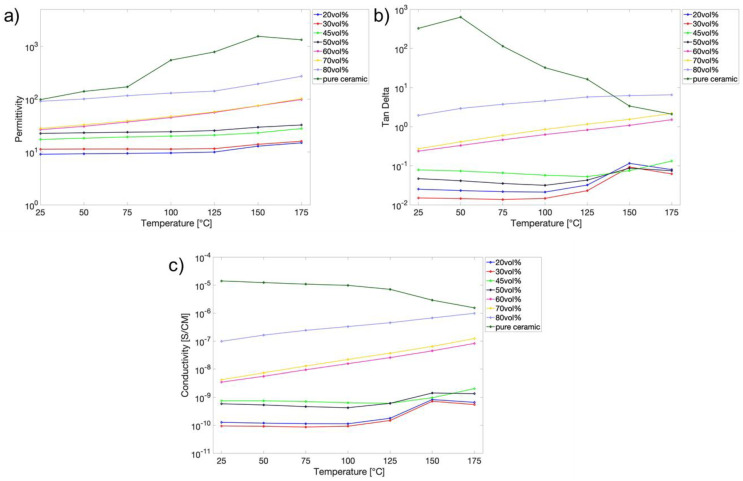
Temperature dependence of: (**a**) permittivity, (**b**) dielectric loss, and (**c**) conductivity of different vol% ZnO varistor-filled composites.

**Figure 6 molecules-27-06067-f006:**
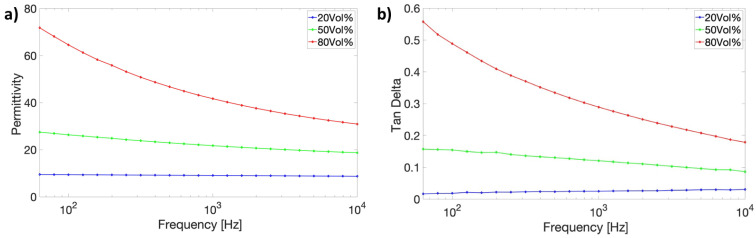
Frequency dependence of (**a**) permittivity and (**b**) dielectric loss of different vol% ZnO varistor-filled composites at room temperature.

**Figure 7 molecules-27-06067-f007:**
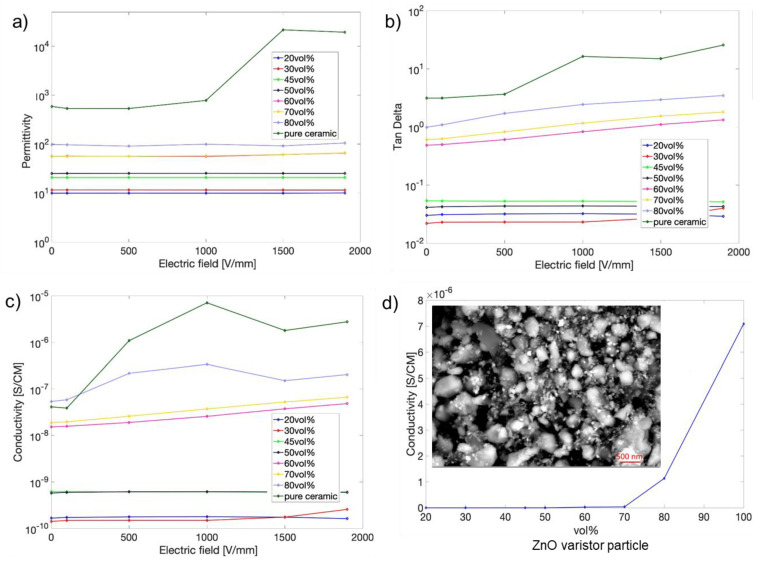
Electric field dependence of (**a**) permittivity, (**b**) loss factor, (**c**) conductivity of the composites filled with different vol% ZnO varistor particles, and (**d**) conductivity as a function of vol% ZnO varistor and the SEM image of the fillers of 80 vol%.

**Figure 8 molecules-27-06067-f008:**
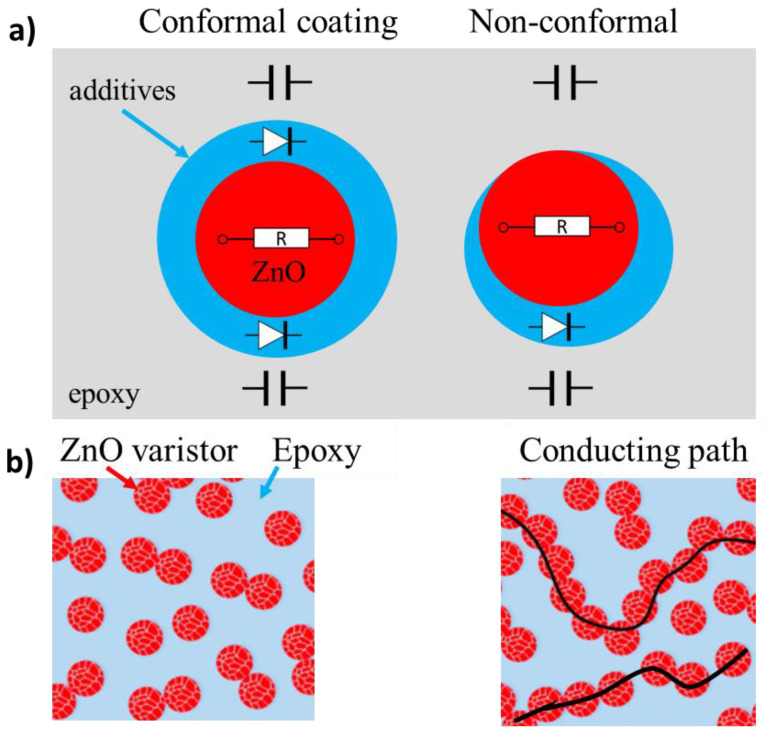
(**a**) The schematic model of ZnO varistor particles in the epoxy matrix. (**b**) The schematic distribution of ZnO varistor particles with isolated dispersion at lower loading (**left**) and interconnected configuration at higher loading (**right**).

**Table 1 molecules-27-06067-t001:** Composition of ZnO Varistor Particles.

Content	ZnO	Bi_2_O_3_	Sb_2_O_3_	MnO_2_	Co_3_O_4_	Cr_2_O_3_	NiO	SiO_2_
wt%	86.08	5.24	4.90	x	y	z	u	v

## Data Availability

The data used to support the findings of this study are already incorporated in the results section.

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
