# Peer review of "Dielectric Loss and Electrical Conductivity Behaviors of Epoxy Composites Containing Semiconducting ZnO Varistor Particles"

_molecules, 2022, doi:10.3390/molecules27186067_

Round 1

Reviewer 1 Report (Previous Reviewer 2)

The paper deals with electrical behavior of epoxy composites containing ZnO particles. The subject is not new, and the innovation of the research should be more clearly emphasized. As long as the literature largely mentions the effect of particle size on the dielectric properties of ZnO particles, and emphasizes the importance of using smaller particles (under 100nm), the use of lager particles - over 500nm - should be explained. The 'varistor' term is largely used in the paper, but poor demonstration on such a behavior of the composites is presented. The purpose of the paper is uncertain, even the conclusions mentioning the best performance of e.g. composites with ZnO particles above 70 vol% must be related to a potential application that needs such electrical effects. The paper mentions that an extremely high dielectric loss (above 1) under higher bias fields would indicate the 'metallic type conducting nature', aspect that is unexpected at un-dopped ZnO particles with metallic particles. Even the conductivity graphic leads to values far from an electronic conducting material behavior. According to literature, the dielectric characteristics of materials with ZnO particles must be performed at frequencies higher than 1kHz, i.e. at least 10kHz in order to better outline the composite effect. In all, the dielectric characteristics are poor, and an analysis vs. vol% may lead to more clear experimental observations. In all - the conclusions must be expanded in order to clear up the purpose of the study, the potential practical applicability and the relevant obtained results, not a simple description of the results based on disputable graphics. 

Author Response

Thank you for your comments. Please see the attachment for details.

Reviewer 2 Report (Previous Reviewer 3)

Minor spell checks are required

Author Response

Thank you!

Round 2

Reviewer 1 Report (Previous Reviewer 2)

The authors fairly addressed the reviewer's observations and completed the paper accordingly. 

This manuscript is a resubmission of an earlier submission. The following is a list of the peer review reports and author responses from that submission.

Round 1

Reviewer 1 Report

The manuscript entitled: "Enhanced Dielectric and Electrical Responses of Epoxy Composites via Impregnated ZnO Core-Shelled Particle Network" is a work dealing with the the fabrication of new electric conductive composites exhibit also enhanced dielectric properties. The manuscript is well written and I cannot identify serious errors or things that need to be changed. I could recommend the acceptance of that after minor revision. However, it does not fit well the scope of this journal. Therefore, I suggest that the manuscript is submitted - transferred to another mdpi journal, such as the journal materials or composites.

As minor scientific comments, I could add:

  • How did the authors calculate the dielectric loss in the case of the foam which also contains free volume?
  • The SEM images of figure 3 need a better visible scale bar
  • Figure 1 please correct the error in the first image regardignthe x axis
  • the EDX in figures 3c and 3d are very small, can you increase the axis size?
  • last but not least, the impregnation is a process that take place under vacuum but the authors do not mention if the impregnation is 100  % or if there is resin that coat the samples on the outer surface. please comment on that.

Reviewer 2 Report

The paper deals with the manufacture and testing of pseudo-composites mainly obtained from core-shell ZnO matrix and epoxy resins. The technology of composites must be more explained and detailed, mainly the one related to ceramic foams - the way of obtaining different porosities, and how those porosities were evaluated. The internal structure of the ceramic foams must be addressed in relation with the impregnation technology, i.e. to what extent the impregnation is considered complete, if closed pores are occurring etc. The used term of 'nanodielectric composites' must be explained. The authors should explain why the electrical features are described for 1kHz only, and why the analysis vs. temperature is relevant at that stage. On the other hand, the analysis of electrical features vs. field (excepting for conductivity) brings no relevant information, but a broadband analysis vs. frequency and temperature would bring more relevant information upon both structure-technology and potential use of such materials. Hypothesis that 'the impregnated foam ceramic method is more effective in exhibiting the filler effect 260 and higher dielectric permittivity than the mixing method that is likely due to the - filler interconnection - pre-set by the foaming agent' must be justified, mainly as the effect seems to be related to the difference in homogeneity of the two technologies, i.e. the ceramic foams may presume o better dispersion of fillers.  Assumption that 'above 70 vol%, the composites are dominated by the nature of ZnO particles which form interconnected conduction paths and give rise to enhanced electric conductivity' - must be justified, in terms of what exactly is meant by 'enhanced electric conductivity' - i.e. what type of conductivity, with some exact comparative values etc. It is obvious that the paper should be expanded to an extent able to better explain both technological stages and assumptions upon the test results, as long as the results presented in the paper seem to be too brief of irrelevant for a pertinent analysis of the obtained structures. The introduction is unclear, with a poor English. 

Reviewer 3 Report

Some of the major concerns are given below.

  1. The core-shell structure of ZnO particles is not evident from the material characterization results provided in the manuscript.
  2. The SEM images in the manuscript don’t evident the nano-sized features of ZnO particles.
  3. The reason for enhanced dielectric permittivity and electrical conductivity in the Epoxy/ZnO-based nanocomposites fabricated by Impregnated Foam Ceramic method compared with the conventional mixing method is unknown.
  4. The nature of molecular interactions between ZnO and Epoxy is not studied.